# The Application of Advanced Information Technologies in Civil Infrastructure Construction and Maintenance

**Clyde Zhengdao Li [1], Zhenchao Guo [1], Dong Su [2], Bing Xiao [3,*] and Vivian W. Y. Tam [3]**

1   Sino-Australia Joint Research Center in BIM and Smart Construction, College of Civil and Transportation Engineering, Shenzhen University, Shenzhen 518060, China; clyde.zhengdao.li@szu.edu.cn (C.Z.L.); 2060474018@email.szu.edu.cn (Z.G.)
2   Key Laboratory for Resilient Infrastructures of Coastal Cities, Ministry of Education, Shenzhen University, Shenzhen 518060, China; sudong@szu.edu.cn
3   School of Engineering, Design and Built Environment, Western Sydney University, Penrith, NSW 2751, Australia; v.tam@westernsydney.edu.au
*   Correspondence: 20260806@student.westernsydney.edu.au

**Abstract:** Information technologies have widely been used in the construction and maintenance of civil infrastructure. The advantages of information technologies provided a broader range of methods for infrastructure and enhanced its level of maintenance. However, a systematic summary of the research development of information technologies used in civil infrastructure is limited. This study aims to supplement this field by providing an objective, systematic summary of relevant literature in mainstream journals employing bibliometric retrieval and quantitative analysis from 2010 to 2020. The following results are obtained: (1) This study discusses the application of advanced information technologies in different phases and provides a critical analysis of the application of these existing information technologies, which includes wireless sensor networks (WSN), fiber optic sensing (FOS), building information modelling (BIM), radio frequency identification (RFID) and other advanced information technologies. (2) The digital twins can be used as tools for the planning and management of next-generation smart infrastructure, making the future of civil infrastructure smarter and more sustainable.

**Keywords:** civil infrastructure; information technology; construction management; literature review

## 1. Introduction

Civil infrastructure includes buildings, roads, bridges, tunnels, pipelines and other transport networks. Although civil infrastructure is designed to last for many years, it may be extended, updated and maintained every decade as society evolves and needs change [1]. As populations continue to grow and infrastructure structures age, there is an urgent need for more effective, cost-effective technologies to build, maintain, monitor and repair these infrastructure structures [2].

With the decrease in the cost of sensor equipment in recent years, the application range of information sensing technology has expanded rapidly. Academic research on the applications of information technologies to design, construct and maintain civil infrastructure in areas such as location optimization [3,4], schedule monitoring [5–8], safety warning [9,10] and structural health monitoring (SHM) is growing [11–16].

Compared with buildings, the majority of resources are spent in the phases after handover of the entire civilian infrastructure life cycle. This finding implies that applications of information technologies are inclined to the maintenance phases in civil infrastructure, especially SHM. SHM is the automatic receipt of infrastructure information data, such as information sensing technology. The information from the sensors is used to detect any abnormal problems that could lead to serious consequences, such as structural damage.

SHM can assist civil engineers in monitoring structural information and making early decisions, which is critical for the reliable operation of infrastructure.

However, there is a lack of systematic analysis and discussion of these information technologies, and an overview of their applications in the full life cycle of infrastructure. This study aims to supplement to this field by providing an objective, systematic summary of relevant literature in mainstream journals employing bibliometric retrieval and quantitative analysis from 2010 to 2020. The information technologies primarily employed in this paper mainly include WSN, FOS, BIM, RFID and other advanced information sensing technologies.

This study provides a thorough and systematic overview of advanced information technology applications in civil infrastructure. Firstly, it introduces the background of information technologies in civil infrastructure and explains the methodology used. The collected papers are then analyzed to determine the information technologies widely used in civil infrastructure and examine their advantages and limitations. Finally, it discusses the current state of information technologies applications in civil infrastructure, the future research directions, and provides the conclusion. This study focuses on civil infrastructure and not the general infrastructure from vertical and horizontal construction. The study also analyzes the information technologies from the perspective of management and application and generally does not involve the analysis of specific technical principles.

## 2. Methodologies

The review methods of previous studies provide a valuable reference for the Selection. Huo et al. [17] provided a bibliometric search method, which defined the search terms according to the literature search objective and explored the database for relevant literature. The literature was classified according to research, content or methodology; and a critical review of existing articles was conducted to demonstrate past progress in the field. Moreover, it is crucial to accurately identify the searched keywords, database, timespan and document type. The research framework of this review is shown in Figure 1.

The research framework of this review is as follows:

(1)   Literature collection and screening

The Web of Science core collection database contains major publications on civil infrastructure, covers a wide range of topics and has a well-known scientific impact. Furthermore, it has more advanced functions, such as citation analysis and retrieval results analysis. Thus, this database was selected for systematic retrieval in this study. The authors searched for a few keywords in the Web of Science core collection database, including 'infrastructure construction', 'infrastructure maintenance', 'information technolog*', 'fiber optics sens*', 'wireless sensor network*', 'micro-electro-mechanical systems' and 'radio frequency identification'. In the advanced search query builder, enter "TS = ((infrastructure construction OR infrastructure maintenance) AND (information technolog* OR fiber optics sens * OR wireless sensor network* OR micro-electro-mechanical systems OR radio frequency identification))", and the search interval was set from 2010 to 2020. Herein, '*' means a fuzzy search. Moreover, journal articles in English have a high global acceptance because they are published after a thorough and rigorous review process; thus, journal articles are more representative than conference papers and books. As a result, SCI-Expanded and SSCI, which are more closely related to this research discipline field, are selected in the Editions of the Web of Science core collection database. A total of 595 literature samples were obtained from the database through the aforementioned Boolean operation.

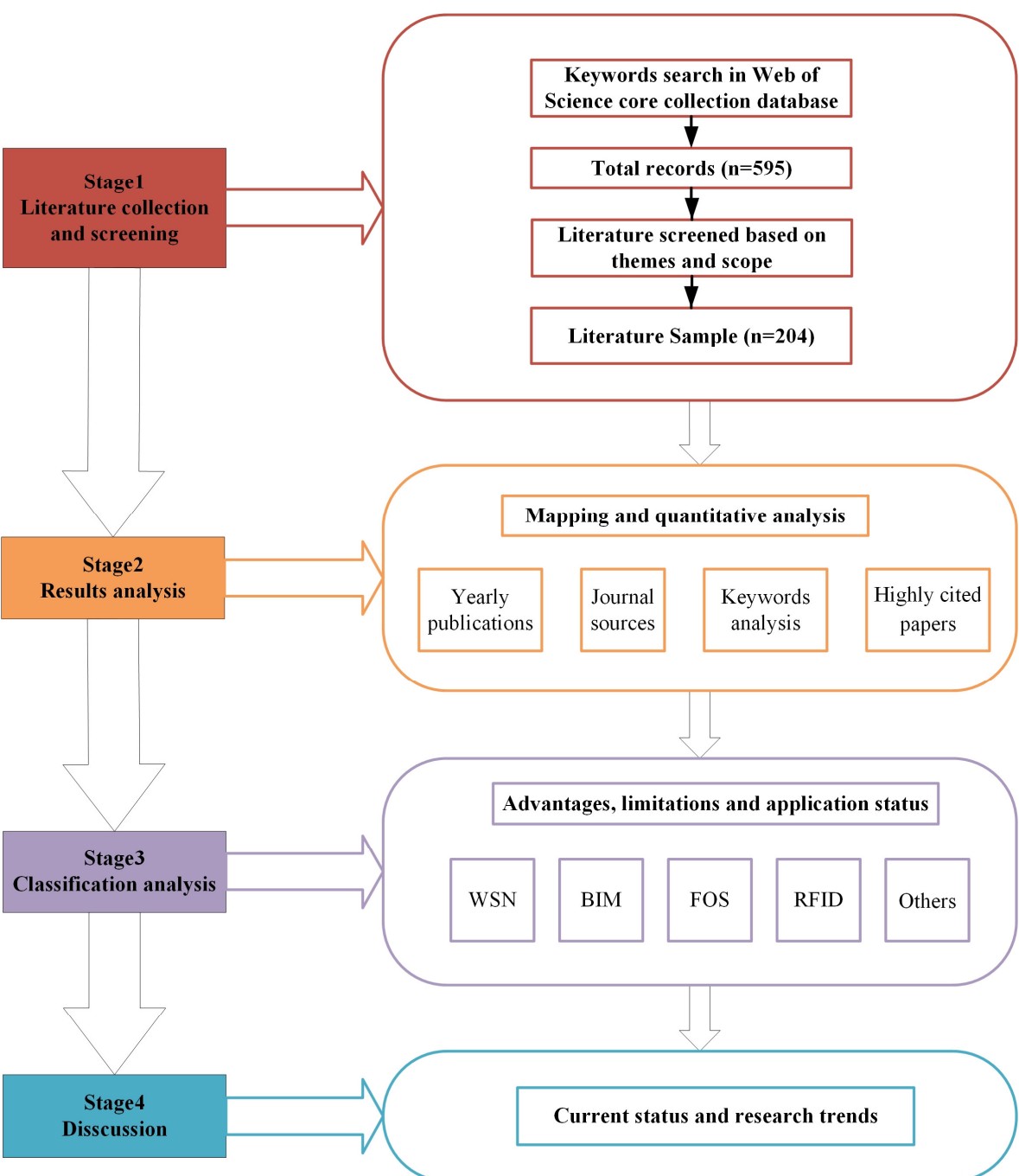

**Figure 1.** The research framework of this review.

Further screening of the aforementioned literature samples according to the following two steps are: (1) to verify the language and type of literature. On average, conference papers provide less comprehensive information and a theoretical framework than journal articles. Therefore, only journal articles or reviews published in English will be retained in the literature samples. (2) to screen out the literature whose research topic and scope are inconsistent with this study. Only literature focusing on the application of information technology in civil infrastructure will be retained. This step eliminates a large amount of literature because the scope of a large portion of the previous literature was not civil infrastructure but rather housing construction or others. Following the above two steps of screening, 204 papers were selected as literature samples for subsequent analysis.

(2)   Analysis of the Results

This study to analyzes and visualizes bibliometric networks through scientific mapping and quantitative analysis. Research trends and journal sources can be determined by using the analytical retrieval results of the Web of Science core collection database. VOSviewer was used to examine the information technology application status of civil infrastructure, including co-occurrence keywords, country activity analysis and highly influential papers.

(3)   Classification analysis

Based on the analysis of the above results, the advanced information technology categories which are widely used in the construction and maintenance of civil infrastructure are determined. The information technologies are then classified and discussed, including their advantages, limitations and application status.

(4)   Finally, based on the above analysis results, the current status and research trends of this research were discovered.

## 3. Results and Analysis

VOSviewer is a literature mining tool, which can be used to classify and analyze the collected literature by keywords, scholars and countries, and can visually display the analysis results. As shown in Figure 2, this study adopted VOSviewer to analyze the bibliometric network relationship of information technologies in infrastructure construction and maintenance, including an annual number of publications, analysis of journal sources, analysis of co-occurrence keywords and analysis of paper citations.

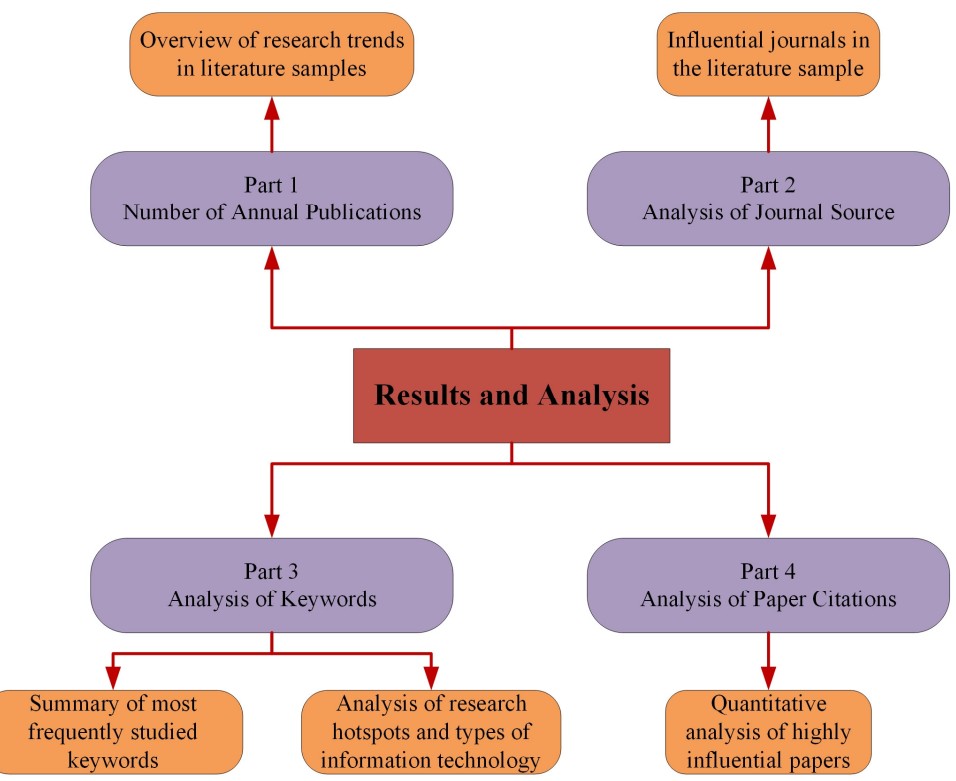

**Figure 2.** The framework of results and analysis.

### 3.1. Number of Annual Publications

Figure 3 shows the classification of the 204 papers after screening according to the year of publication. The number of papers in this field is generally increasing. The solid line represents the number of papers published each year, whiles the dotted line shows

an increase in the number of papers published in recent years. This trend demonstrates that academics have become increasingly interested in the application of information technologies in infrastructure.

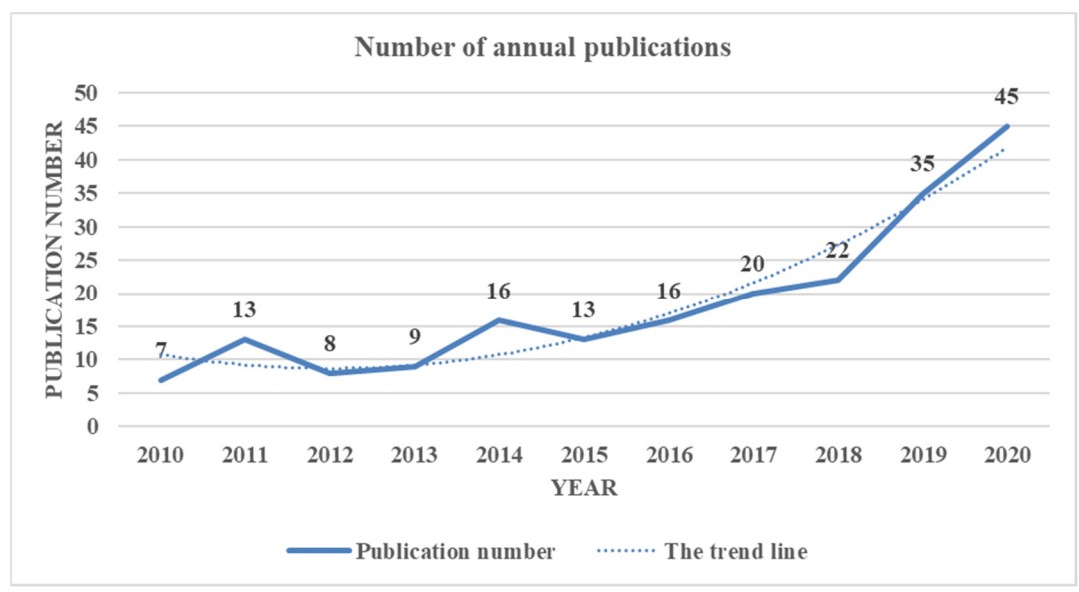

**Figure 3.** Number of annual publications from 2010 to 2020.

### 3.2. Analysis of Journal Source

Table 1 lists the top nine journals in this research based on the number of papers published, including 'Sensors', 'Automation in Construction', 'Smart Structures and Systems', 'Journal of Civil Structural Health Monitoring', 'Applied Sciences-Basel', 'International Journal of Distributed Sensor Networks', 'Structural Control and Health Monitoring', 'Structure and Infrastructure Engineering' and 'Sustainability'. Amongst them, 'Sensor' was the journal with the largest number of published papers, a total of 26, and 'Automation in Construction' had the most citations, a total of 268. Furthermore, 'Structural Control and Health Monitoring' was the most cited journal on average, with 29.8 citations per paper. However, high-quality journals and articles published in relatively small quantities will be discussed in subsequent chapters.

**Table 1.** Top nine journals in this research by the number of papers published.

| No. | Journals | Number of Papers | Total Citations | Norm. Citations | Avg. Citations | Avg. Norm. Citations | Avg. Pub. Year |
|-----|----------|------------------|-----------------|-----------------|----------------|----------------------|----------------|
| 1 | Sensors | 26 | 249 | 25.58 | 9.58 | 0.98 | 2018 |
| 2 | Automation in Construction | 16 | 268 | 25.22 | 16.75 | 1.58 | 2017 |
| 3 | Smart Structures and Systems | 7 | 162 | 10.50 | 23.14 | 1.50 | 2015 |
| 4 | Journal of Civil Structural Health Monitoring | 6 | 32 | 1.60 | 5.33 | 0.27 | 2018 |
| 5 | Structural Control and Health Monitoring | 5 | 149 | 11.64 | 29.80 | 2.33 | 2014 |
| 6 | Structure and Infrastructure Engineering | 5 | 50 | 2.60 | 10.00 | 0.52 | 2015 |
| 7 | International Journal of Distributed Sensor Networks | 5 | 20 | 1.68 | 4.00 | 0.34 | 2014 |
| 8 | Sustainability | 5 | 6 | 1.57 | 1.20 | 0.31 | 2019 |
| 9 | Applied Sciences-Basel | 5 | 8 | 0.85 | 1.60 | 0.17 | 2019 |

### 3.3. Analysis of Keywords

Using 'All Keywords' to analyze the keywords in VOSviewer, the minimum frequency of occurrence was set as 7, and integrate the Keywords that represent the same meaning, such as 'building information modelling' versus 'building information model' and 'fiber optic sensing' versus 'fiber optic sensors'. Initially, 27 of the 1063 keywords met this requirement. Then, several vague or inclusive terms, such as 'system', 'framework' and 'classification' were eliminated. Moreover, the following chapters were specifically introduced because sensors and the Internet of things (IoT) are the foundations of many information sensing technologies. Finally, Table 2 and Figure 4 show the analysis of the 20 keywords.

**Table 2.** Top 20 keywords in this research by the number of occurrences.

| No. | Keywords | Occurrences | Total Link Strength | Avg. Citations | Avg. Norm. Citations | Avg. Pub. Year |
|---|---|---|---|---|---|---|
| 1 | Structural health monitoring | 44 | 93 | 20.41 | 1.56 | 2016 |
| 2 | Wireless sensor networks | 43 | 79 | 19.84 | 1.62 | 2017 |
| 3 | Information technology | 28 | 77 | 10.32 | 0.81 | 2017 |
| 4 | BIM | 27 | 73 | 16.81 | 1.09 | 2018 |
| 5 | Fiber optics sensing | 23 | 44 | 16.70 | 1.06 | 2016 |
| 6 | Management | 19 | 62 | 10.11 | 0.82 | 2017 |
| 7 | RFID | 19 | 62 | 19.05 | 1.13 | 2015 |
| 8 | Design | 18 | 62 | 20.83 | 1.28 | 2017 |
| 9 | Infrastructure | 17 | 54 | 11.82 | 0.80 | 2017 |
| 10 | Construction | 16 | 60 | 18.50 | 1.19 | 2017 |
| 11 | Bridges | 14 | 35 | 17.29 | 1.28 | 2016 |
| 12 | Visualization | 11 | 39 | 11.73 | 0.76 | 2018 |
| 13 | Crack detection | 10 | 24 | 9.70 | 1.17 | 2018 |
| 14 | Damage detection | 10 | 31 | 23.70 | 1.71 | 2017 |
| 15 | Internet of things | 10 | 25 | 6.50 | 1.06 | 2019 |
| 16 | Inspection | 9 | 30 | 19.22 | 1.85 | 2016 |
| 17 | GIS | 8 | 20 | 12.63 | 0.65 | 2018 |
| 18 | Maintenance | 8 | 22 | 22.38 | 1.13 | 2016 |
| 19 | Algorithm | 7 | 16 | 30.86 | 2.85 | 2017 |
| 20 | Tracking | 7 | 23 | 27.86 | 1.65 | 2015 |

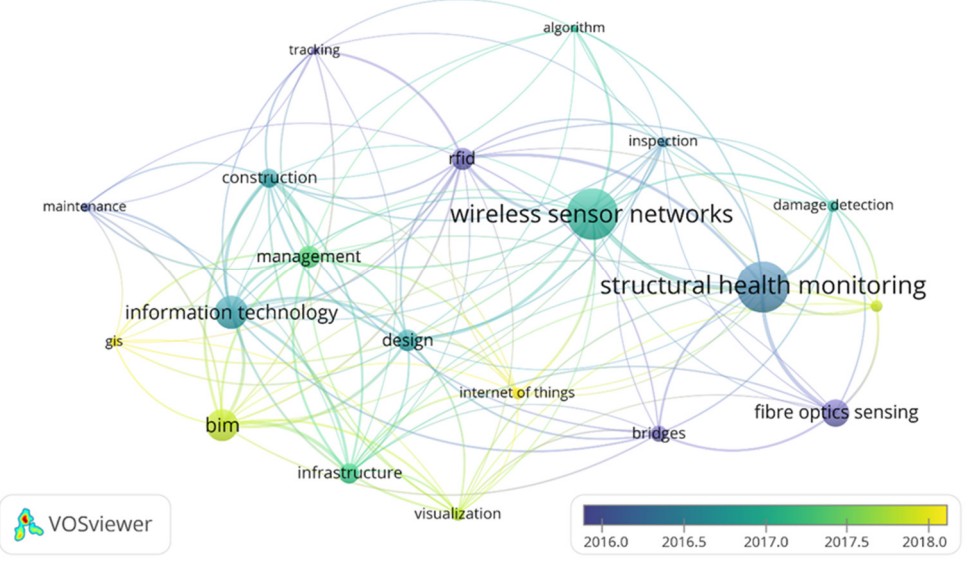

**Figure 4.** Co-occurrence of keywords.

According to Table 2, the following information was found:

Firstly, two indicators were involved; 'Occurrences' refers to the times of the keywords, and 'Total Link Strength' refers to the total number of co-occurrences (including repeated co-occurrences) between a keyword and other keywords. The keywords of 'structural health monitoring' appeared the most, 44 times. Therefore, SHM is the most popular research direction in this field. Secondly, WSN, BIM, FOS, and RFID were the four most frequently used information technologies in infrastructure construction and maintenance. The following chapters will expound on the detailed introduction.

Figure 4 shows the classification of the time of occurrence of keywords in VOSviewer, RFID and FOS was applied before 2016, whereas BIM, internet of things (IoT) and geographic information system (GIS) are the research hotspots after 2018.

### 3.4. Analysis of Paper Citations

The minimum number of citations was set to 50, and 11 out of 204 papers met the threshold as shown in Table 3. Highly cited papers and authors in the field can be identified. In addition, more than half of the papers obtained are review papers, and the research topics focus on the SHM field.

**Table 3.** Highly cited papers in this research.

| No. | Author | Title | Journal | Total Citations | Norm. Citations |
|-----|--------|-------|---------|-----------------|-----------------|
| 1 | Hodge et al. [18] | Wireless Sensor Networks for Condition Monitoring in the Railway Industry: A Survey | IEEE Transactions on Intelligent Transportation Systems | 172 | 5.47 |
| 2 | Liu et al. [7] | A Review of Rotorcraft Unmanned Aerial Vehicle (UAV) Developments and Applications in Civil Engineering | Smart Structures and Systems | 114 | 6.12 |
| 3 | Noel et al. [19] | Structural Health Monitoring Using Wireless Sensor Networks: A Comprehensive Survey | IEEE Communications Surveys and Tutorials | 107 | 5.66 |
| 4 | Torres et al. [20] | Analysis of the Strain Transfer in a New FBG Sensor for Structural Health Monitoring | Engineering Structures | 74 | 2.61 |
| 5 | Bocca et al. [21] | A Synchronized Wireless Sensor Network for Experimental Modal Analysis in Structural Health Monitoring | Computer-Aided Civil and Infrastructure Engineering | 72 | 2.54 |
| 6 | Liu et al. [22] | A State-of-the-Art Review on the Integration of Building Information Modelling (BIM) and Geographic Information System (GIS) | ISPRS International Journal of Geo-Information | 70 | 3.70 |
| 7 | Mottola et al. [23] | Not all Wireless Sensor Networks are Created Equal: A Comparative Study on Tunnels | ACM Transactions on Sensor Networks | 69 | 2.98 |
| 8 | Park et al. [10] | Performance Test of Wireless Technologies for Personnel and Equipment Proximity Sensing in Work Zones | Journal of Construction Engineering and Management | 66 | 3.65 |

**Table 3.** *Cont.*

| No. | Author | Title | Journal | Total Citations | Norm. Citations |
|-----|--------|-------|---------|-----------------|-----------------|
| 9 | Dai et al. [8] | Comparison of Image-Based and Time-of-Flight-Based Technologies for Three-Dimensional Reconstruction of Infrastructure | Journal of Construction Engineering and Management | 59 | 4.18 |
| 10 | Teizer et al. [24] | Status Quo and Open Challenges in Vision-based Sensing and Tracking of Temporary Resources on Infrastructure Construction Sites | Advanced Engineering Informatics | 56 | 1.78 |
| 11 | Sony et al. [25] | A Literature Review of Next-generation Smart Sensing Technology in Structural Health Monitoring | Structural Control & Health Monitoring | 52 | 6.77 |

## 4. Classified Analysis of Information Technologies in Civil Infrastructure

According to the keyword analysis results, the four most frequently used information technologies in civil infrastructure construction and maintenance are WSN, BIM, FOS and RFID. Therefore, the application status of these four information technologies will be discussed in the following chapters. Sensors and IoT, computer vision (CV), GIS and micro-electro-mechanical system (MEMS) involved in civil infrastructure will also be discussed.

### 4.1. Wireless Sensor Networks

WSN has emerged as a powerful, low-cost platform for connecting large networks of sensors. WSN refers to the collection of and transmission of information by sensors and sensor data using wireless sensor technology, which can be used to monitor infrastructure. WSN consists of microcontrollers, sensors and radios, all of which require low power consumption. Figure 5 shows a framework of WSN applications in civil infrastructure.

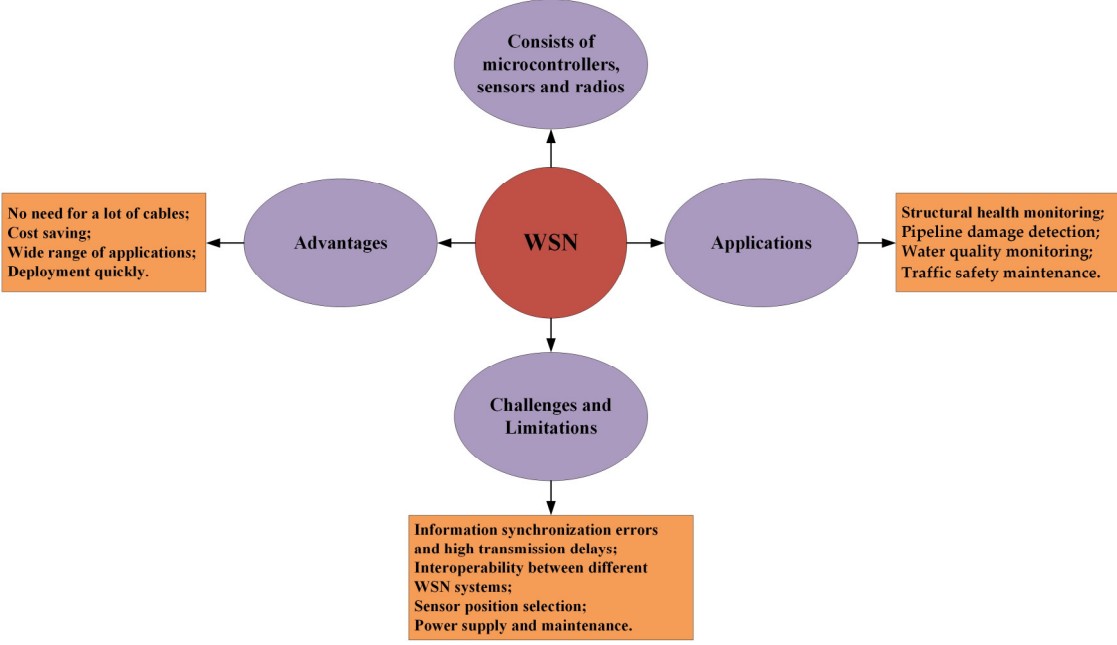

**Figure 5.** A framework of WSN applications in civil infrastructure.

### 4.1.1. Advantages

Compared with current wired sensors and sensing systems, the advantages of WSN includes; allowing the deployment of a large number of sensors without the need for cables, resulting in significant cost savings, can be used to monitor areas that are difficult to wire and can be deployed quickly to allow stakeholders make earlier engineering decisions compared to wired deployments [1].

### 4.1.2. Applications

In recent years, the deployment of WSN in the field of civil infrastructure has gradually become a hot research direction, including the detection of pipeline leakage, monitoring of water quality, control of vehicle overloading, monitoring of roads, monitoring of railway infrastructure and vehicle health, especially in the field of SHM.

#### Structural Health Monitoring

WSN offers numerous advantages for SHM such as robust data management, increased higher flexibility and lower costs [26]. WSN for SHM sensors are deployed at different points throughout the structure to collect data, such as acceleration, vibration, load and stress. WSN-based SHM systems have been widely used to monitor infrastructures such as bridges, buildings and sports venues; it reduces the costs of installing and maintaining SHM systems. Also, they have the potential to improve structural life and public safety.

#### Pipeline Damage Detection

Noel et al. [19] presented an investigation of SHM using WSN and outlined the algorithms used for damage detection and localization. Utility tunnel is one of the important infrastructures in smart city planning and construction. Pipeline leakages may result in large excessive costs and serious environmental pollution [13]. WSN can be used to improve the efficiency of maintenance and overhaul of urban pipelines [12]. A key challenge for SHM systems in underground pipeline monitoring is that when the wireless device is buried in the ground, the wireless signal is significantly weakened during transmission. Ali et al. [27] proposed a buried wireless sensor network (B-WSN) system to detect pipe joint leakage caused by large-scale ground movement. The B-WSN system was equipped with a high-energy signal transceiver; thus, each sensing node also acts as a relay node to assist in data communication with nodes buried deeper underground.

#### Water Quality Monitoring

Water resources are vital in our daily life, Imran et al. [28] developed an autonomous water quality monitoring system based on a smart city WSN. WSN was used to obtain good quality observations, including PH, turbidity, dissolved oxygen and temperature.

#### Traffic Safety Maintenance

Hodge et al. [18] conducted a survey that demonstrated how WSN can be used to monitor transportation infrastructures such as bridges and tracks, and the health of vehicles in real-time. Zhao et al. [29] developed a piezoelectric sensing system for road surfaces to control the adverse effects of vehicle overload on traffic safety. Furkan et al. [30] developed a WSN system derived from wired sensors for the fast, robust measurement of structural deformation on highways. WSN has been widely used in the railway industry.

### 4.1.3. Challenges and Limitations

At present, the application of WSN in the field of infrastructure still faces many challenges.

Firstly, sensor node density may be high, and the number of hops from the node to the base station may be large because WSN collects, processes and transmits a large amount of data [19]. These situations may result in synchronization error of information between sensor nodes or a high delay in information transmission. WSN systems require complex algorithms to ensure data correctness and transmission stability. They also need

the merging of data from other sensor nodes and usually require centralized processing. Meanwhile, interoperability between different WSN systems is extremely limited [1].

Secondly, the location of the sensor determines its efficiency in collecting information. If the selected sensor is poorly located, the information collected limits the system's ability to detect, locate damage and reduces the life of the WSN system.

Finally, the energy of each sensor node in the WSN system is limited; thus, frequently replacing the battery takes a large amount of time and cost. The capacity of the WSN system needed to shut down unnecessary nodes to save energy without detecting an event is a major concern [31]. Although placing the sensor nodes in a better position can improve work efficiency and extend the life of the WSN system, the maintenance method of the WSN system is an important challenge to mitigate node energy consumption in large-scale applications. To some extent, current wireless sensor systems still rely on wired connections for communication and power supply to some extent [30].

### 4.2. Building Information Modelling

BIM has been widely used in the construction industry for decades, but its applications in civil infrastructure projects progressing slowly [32]. In recent times, industry and academia have been collaborating to apply BIM for civil infrastructure projects, such as roads, highways and bridges. BIM is a decision-making tool that utilizes various digital tools and applications [33]. BIM provides construction professionals with new methods to plan, design, build and manage infrastructure more effectively [34].

### 4.2.1. Advantages

3D Visualization

One of the most obvious advantages of BIM is its ability to convert 2D structural engineering drawings into 3D models and visualize large amounts of structural-related data and building components. This automated tool provides better system maintenance and risk management whilst avoiding human error due to visual inspection of the structure [35]. Eleftherakis et al. [36] introduced a secondary development system of underground pipeline 3D modelling based on BIM. This system can model the underground pipeline network and visualize it in 3D using BIM technology.

Improve the Efficiency and Accuracy of Information Exchange

BIM relies on a variety of digital technologies and can be used to model information about civilian infrastructure. The data in the BIM model is characterized by objectivity, applicability, mobility and share-ability, and the efficient information exchange ensures the integrity and accuracy of information and data exchange [37].

Life Cycle Management

BIM can be implemented to enhance life cycle information management from the initial planning and design phase to the final demolition stage of a project. The BIM model can be used to obtain detailed information about the facility, forming a reliable basis for decision making throughout the life cycle of the facility. It can create, manage and maintain all critical information related to assets in the design, construction and maintenance of road or rail infrastructure projects [38]. Bridge information modelling, for example, is an intelligent representation of a bridge and contains all the bridge information needed throughout the life cycle of the bridge. It can collect and store information during the life cycle of the bridge, which can effectively help bridge maintenance in the future, reduce maintenance costs and improve bridge safety, quality and efficiency [2,39].

Integrating Emerging Technologies

Emerging technologies such as RFID, laser scanning, mobile computing and cloud computing can be integrated into the BIM platform to improve project performance.

4.2.2. Applications

Based on the numerous advantages of BIM technology, the benefits of BIM application in the field of civil infrastructure are summarized from the perspective of the life cycle, as shown in Figure 6.

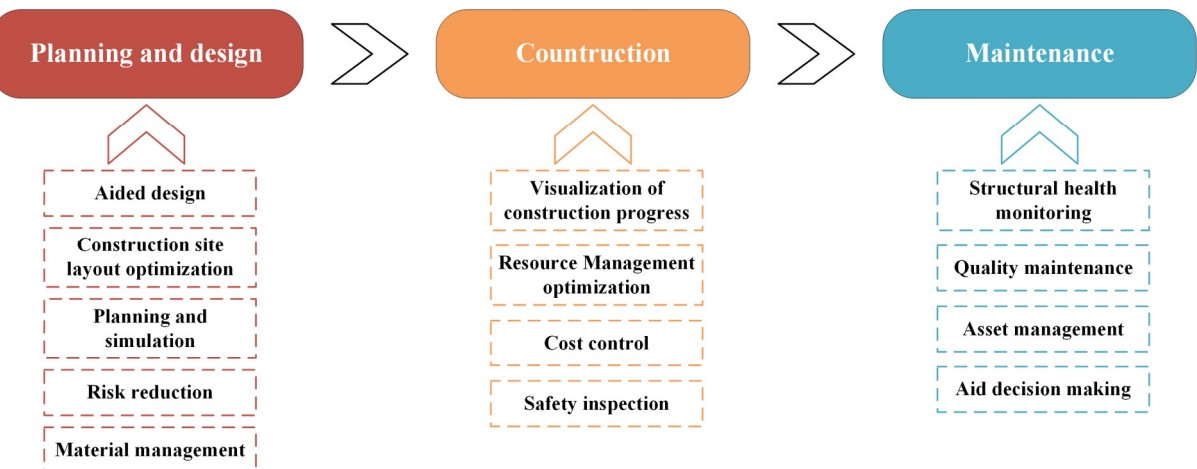

**Figure 6.** A summary of BIM applications in the life cycle of civil infrastructure.

Planning and Design Phase

Aided design: BIM can be useful at different stages of design, including preliminary design, detailed design and design optimization. It also improves design quality through visualization and increases collaboration among various professionals [40].

Construction site layout optimization: Generally, the construction of civil infrastructure has different locations and paths. BIM can be combined with other information-sensing technologies to provide visual, qualitative and quantitative information to help optimize the location of the facility, thereby improving accessibility and operability during the construction and maintenance phases.

Planning and simulation: BIM models created from existing site plans and construction drawings can assist construction participants in providing accurate information and coordinates required for site setting. BIM models can serve as an excellent platform for planning and simulation. With BIM and intelligent algorithms, the layout of the site can be automatically organized and planned with the careful participation of engineers in the planning and design phase [41].

Risk reduction: BIM helps builders in reducing trial and errors during the construction phase, increasing productivity while decreasing risks associated with time and cost [2]. The application of BIM in civil infrastructure from the design phase can help reduce (a) omissions and errors in engineering information (b) conflict and coordination issues on the construction site, and (c) waste. It also improves construction quality and promotes sustainable construction and development.

Material management: BIM models can be integrated with global positioning systems, RFID or GIS technologies to track and manage the status of materials, improving the efficiency of materials supply and management [42].

Project scheduling: Users can visualize the progression of construction at a predetermined time scale by using 4D simulation from BIM models. This will improve the schedule resourcing, monitoring, and updating processes.

Construction Phase

Visualization of construction progress: The data captured by video monitoring and transmitted back to the BIM model can be used to track the construction progress on site. It allows for more effective monitoring and control of field progress. Lei et al. [5] proposed

a data-driven deep learning algorithm based on a convolutional neural network, which combines a 3D point cloud with a building information model and uses it to identify and visualize work progress with high accuracy.

Resource Management optimization: BIM can reasonably arrange resource supply based on the actual progress of the project, plan raw material demand in advance and reduce resource waste. It can increase productivity by improving the efficiency of resource utilization and preventing waste of time and resources during the construction process, thereby speeding up the construction process [43].

Cost control: 5D BIM incorporates cost-related latitude into the model, allowing project stakeholders to track and update cash flow information [44]. Changes to the model will be reflected in the cash flow information accordingly.

Safety inspection: Liu et al. [45] proposed an efficient safety inspection method, which combined unmanned aerial vehicle (UAV) and BIM to realize synchronous navigation to improve the efficiency and accuracy of safety inspection. It can also be used to inspect other large infrastructures, such as sidewalks, dams and railroads.

Maintenance Phase

Structural health monitoring: BIM enables visual monitoring and analysis of large amounts of sensor data and structural health information over long periods and state assessment from long-term monitoring data [35].

Quality maintenance: BIM creates a centralized system for visualizing quality maintenance, which can improve structural performance and reduce life cycle costs [46].

Asset management: BIM technology can assist managers in visualizing asset information management. This includes visual asset monitoring, querying and location management through a combination of BIM and related technologies. Additionally, effective facility management can increase project value and extend the project life.

Aid decision making: BIM and related digital innovations can promote greater transparency, tighter integration and higher productivity in the construction industry. Hu et al. [47] proposed an electronic maintenance framework that combines BIM and Semantic Web technologies to help integrate heterogeneous data and expertise, enabling lifelong information sharing, and supporting managers to make effective maintenance decisions.

### 4.2.3. Challenges and Limitations
Application of Obstacles

Researchers surveyed 200- active construction companies to assess the impact of barriers on the adoption of BIM. According to the findings, factors influencing the failure to adopt BIM include lack of trained personnel, appropriate social infrastructure, guidance and government support [48]. The purpose of BIM is to obtain all information about the design and construction of facilities and to use it for subsequent maintenance of facilities, but the application of BIM is currently mainly concentrated in the early design phase.

Need for Interoperability

Flora et al. [49] believed that BIM alone is not sufficient to achieve the ambitious goal of optimizing building planning and subsequent management. Interoperability has proven to be a major barrier to the seamless transfer of heterogeneous devices. For example, a passive RFID system is integrated into a BIM model for real-time visualization of real-time data from locations in the BIM model, which cannot be achieved without interoperability [50]. The integration of BIM and geographic information system (GIS) can be used for site layout and measurement [32]. However, many integration challenges exist because GIS and BIM were originally developed for different purposes [22].

A fully interoperable system should guarantee no errors, omissions, or data loss when information is transferred from one application to another [2]. Existing models contain many inconsistencies and ambiguities that can hinder successful information exchange.

*4.3. Fiber Optic Sensing*

The SHM of infrastructures is one of the essential modern services for civil engineering structures. SHM systems are designed to continuously evaluate structural integrity, thereby improving safety and optimizing operational and maintenance costs. In recent years, fiber optic sensing technology has become a popular method for predicting structural damage during its service life [51,52]. Changes in the environmental parameters of the structure can be monitored by attaching the cable to the structure or embedding it inside the structure. Fiber optic sensors expand or contract when the structure experiences strain or temperature changes. According to the change in fiber length, the optical signal is reflected back to the analysis unit, and then the relevant data about the structure is obtained [53]. FOS technology is usually used in the measurement and monitoring of strain, temperature, acceleration, deflection, displacement, crack, vibration and corrosion of structures.

### 4.3.1. Advantages

Fiber optic sensors have many distinct advantages compared with conventional sensors, including smaller size and weight, higher accuracy and improved durability and embedding capability [52]. Fiber optic sensors typically cost less than traditional point-measuring sensors. They eliminate the need for copper power cables or battery maintenance and are easy to install.

### 4.3.2. Classification

In this research, fiber optic sensing technologies are mainly divided into two categories, distributed fiber optic sensing (DFOS) systems based on Brillouin optical time-domain reflectometry (BOTDR) and fiber Bragg grating (FBG) sensor systems. Table 4 shows the comparative analysis of DFOS and FBG.

**Table 4.** Comparative analysis of DFOS and FBG.

| Classification | Advantages | Applications |
|---|---|---|
| DFOS | capable of measuring continuous strain and temperature over a long distance; Lower costs | long distance or large range measurements |
| FBG | linear, small size, high resolution and automatic signal transmission | localized measurement |

### Distributed Fiber Optic Sensing

DFOS system based on BOTDR can measure and monitor the strain and temperature generated along with the fiber. It is ideal for monitoring the health of large civil infrastructure because it can perform real-time measurements with high accuracy and stability [54]. The DFOS system based on BOTDR provides continuous, distributed strain measurements that enable civil engineers to understand the stresses and strains generated within the structure. Its main advantages are high sensitivity over long distances and its ability to interact with various objects in a distributed manner.

### Applications

Soga et al. [1,55] discussed the principle of measurement and its latest status in terms of functions. DFOS system based on BOTDR has high resolution and precision, and experiments prove that the DFOS system works well on concrete materials, with the measured strain revealing directional cracks [14]. Bassil et al. [56] evaluated the DFOS technology's ability to locate cracks and quantify crack openings, and compared it with standard conventional sensors. The results showed that DFOS technology can detect early subtle changes in reinforced concrete structures and realize more effective monitoring of reinforced concrete structures with early detection and localization to quantify crack cracking. He et al. [57] proposed a distributed geopolymer-fiber optic sensing system that can detect excessive strain events in steel and measure crack sizes in concrete. Li et al. [58]

investigated the variation of strain conditions in monitoring the key stages of construction in the station site project. The measurement results of DFOS were consistent with those of the conventional inclinometer method, which proved its feasibility in monitoring underground earthwork infrastructure. Di Murro et al. [59] used the DFOS system to measure eight tunnel sections and monitor the long-term deformation of the concrete lining in tunnel. Fan et al. [60] proposed an intelligent concrete equipped with a distributed fiber optic sensor that can use sensor data to monitor in situ the expansion and mass loss caused by steel reinforcement corrosion in steel fiber-reinforced concrete.

Fiber Bragg Grating

SHM for civil infrastructure using fiber Bragg grating sensor networks (FBGSN) has received extensive public attention in recent years [61]. The FBG sensor is a kind of optical fiber sensor with a changing refractive index in the fiber core. Ye et al. [53] elaborated on the measurement principle of FBG sensor. The advantages of using FBG sensors for continuous monitoring are far superior to strain gauges and accelerometers due to the advantages of FBG sensors, which include their small size, lightweight, lack of electrical conductivity, higher accuracy, corrosion resistance, greater durability and embedding capability [52,53,61]. FBG sensors can be embedded inside the structure and fixed to the surface of the structure being monitored [20].

Applications

Liu et al. [61] used FBG sensor technology to design and deploy the SHM system at Tianjin Port, China. FBG blot sensors, FBG temperature sensors, FBG tilt sensors, FBG pressure sensors and FBG acceleration sensors were installed. The results showed that the FBG sensors are very suitable for large-scale, continuous deployment in the high pile wharf. To explore the mechanism of slope instability caused by seepage, Eleftherakis et al. [52] embedded nine series of FBG strain sensors into the soil masses and recorded the vertical and horizontal strain sensing data in the model. The results showed that the FBG strain sensor accurately reflected the gradually accumulated deformation of the slope model under seepage. Wan et al. [62] studied the microstrain distribution obtained from distributed long-gauge FBG sensors for bridge evaluation and health monitoring. The test findings revealed that the distributed long-gauge FBG sensing technology can obtain not only information such as the deflection of the bridge but also parameters such as strain to detect the damage to the bridge. Rodrigues et al. [63] also developed a displacement sensor based on FBG technology to measure the deflection of bridges. The results showed that this sensor can be used in temporary and permanent SHM systems with automatic and remote data acquisition capabilities. Antunes et al. [64] reported monitoring adobe masonry structures using FBG accelerometers and FBG displacement sensor networks with a maximum relative error of 2.08% compared with electronic accelerators and seismographs. The results show that the optical sensor can be used to monitor the health of large original soil and rock structures.

In addition, Zhou et al. [65] developed a combined DFOS and FBG system to meet the requirement of simultaneously locating the full range of structural damage and precise local damage details. The integrated system can simultaneously provide accurate strain measurements at key points by FBG sensors while providing crude distributed strain measurements by DFOS sensors.

*4.4. Radio Frequency Identification*

RFID can be used for contactless two-way data communication through radio frequency mode and can be used to capture or transmit data from tags [66]. Generally, an RFID system consists of an RFID tag and an RFID reader [9]. The last decades have witnessed a rapid growth of RFID technology for identification and tracking because of its unique identification system.

### 4.4.1. Advantages

The traditional manual data collection is time-consuming and laborious. RFID-based equipment can automatically identify and track building materials, providing managers with timely and accurate material information [66]. RFID tags, unlike traditional barcodes, have a unique identification number and can respond to radio waves. As a result, RFID technology is not limited to line of sight such as bar codes and typically has a range of communications over a distance, making it simple to automate identification in harsh environments, such as building and facility operations [67].

### 4.4.2. Applications

#### Proximity Monitoring and Safety Warning

Jo et al. [9] proposed an RFID based proximity warning system to alert workers and equipment operators of potentially hazardous proximity situations because many deaths and injuries occur at construction sites due to workers colliding with objects or equipment. In addition, when RFID sensors detect proximity to workers, the motherboard immediately shuts down the excavator to prevent accidents. Park et al. [10] evaluated commercially available RFID-based proximity monitoring and alarm systems and found them reliable with a very low rate of false alarms.

#### Tracking and Monitoring

Sardroud et al. [66] studied a new method that integrates RFID and a global positioning system (GPS) for real-time data collection in buildings. This method uniquely identifies materials, components and equipment in a way that that allows them to be located and tracked in three phases: production site, in transit and construction site. Akanmu et al. [68] also developed an integration of BIM, genetic algorithms and RFID systems that can report the location of objects in real-time.

#### Pipeline Monitoring and Maintenance

The existing pipeline network active surveillance and frequent inspection method is not only expensive but inefficient because pipelines can be installed in a large-scale complex environment, such as the underground. To overcome these challenges, Kim et al. [69] developed a system based on RFID pipeline automatic maintenance that is efficient and accurate in the inspection of pipeline and maintenance, while providing accurate location information. Almazyad et al. [70] also introduced a water pipeline leakage monitoring system using RFID and WSN technology, to monitor the abnormal situation such as water pipeline leakage and burst.

### 4.4.3. Limitations

The use of RFID tags may be limited if there is a lot of metal and water in the environment [67]. In addition, the RFID system generally uses electromagnetic waves as the transmission medium and space as the transmission channel. Due to a large number of tags, the response signals from different tags may interfere with each other after the reader sends signals. In practical application, combining RFID technology with other technologies is usually necessary for visual monitoring.

### 4.5. Other Advanced Information Technologies

In addition to the above four information technologies, this study involves several other advanced information sensing technologies, such as IoT and sensors, computer vision, GIS, AR (Augmented Reality) and VR (Virtual Reality).

### 4.5.1. IoT and Sensors

IoT literally means 'things' connected to the Internet (for example, sensors and other smart devices) [71]. IoT technologies will play a key role in achieving smart and sustainable urban infrastructure. Sensors, information and communication technologies are

the foundation of IoT devices [50]. Smart sensors and sensing technologies are used to collect, analyze and store data, as well as extract information that can be used to monitor urban infrastructure, diagnose faults and control complex urban systems [72]. In this study, various sensors are involved to monitor the infrastructure.

Concrete Structure Monitoring

New buildings and aging infrastructure need to be monitored for load and damage. Wolf et al. [73] proposed a new ultrasonic transmission sensor and data processing method for that can be used for the non-destructive monitoring of concrete and can be installed during or after construction. D'Alessandro et al. [74] presented a study in which cement-based sensors doped with carbon nanotubes were used to monitor the strain of reinforced concrete. These sensors can be used to monitor structures during their lifetime because they are made of structural materials.

Road Surface Monitoring

A smooth road surface improves driver safety and comfort. Traditional road surface monitoring methods based on vision rely on certain environmental conditions, such as lighting and shadow influence. The development of sensor systems and information communication technology has promoted the development of new road detection methods [75]. In the past few years, smartphone-based sensing has become an important complementary technology to detect road anomalies and monitor potholes, cracks and bumps in the road [76–78]. In addition, traffic conditions can be monitored using a smartphone's microphone sensor, which does not require additional hardware costs; as every time the mobile device is opened can become an active source of information [79]. Based on the temperature sensor, Godoy et al. [75] designed a road monitoring system in which the nodes were temperature probes embedded in the road surface the for real-time monitoring of road temperature.

Other Application Areas

Perez-Padillo et al. [80] developed an intelligent pressure monitoring and alarm system based on sensors that measure hydraulic variables. The IoT platform was used for data analysis and visualization, and it is capable of detecting water supply pipe failures and leaks in real-time. Zymelka et al. [81] presented a new concept of a thin-film strain sensor that measures the local strain distribution and indicates areas with a risk of crack formation.

4.5.2. Computer Vision

CV refers to the use of sensing technology such as cameras or machine vision instead of human eyes for target recognition, tracking, measurement, and further graphics processing. CV can monitor not only the construction progress but also the structural health of the infrastructure in real-time. Collecting accurate, complete and reliable field data is critical to the management of the project site and the entire life cycle [67]. Compared with traditional manual inspection, CV is more efficient and accurate, and can make quick, easy measurements without interrupting construction or operation [82]. The CV technology involved in this study mainly includes UAV, photogrammetry, point cloud, laser scanning and infrared thermal imaging; these technologies are frequently used in conjunction in the field of infrastructure monitoring. Generally, using CV for infrastructure monitoring requires three steps. The first step is data collection, that is, the use of cameras, laser scanning and other sensing technologies to collect structural information; the second step is data processing and analysis, that is, using algorithms and other tools to process the obtained image and video information [24]. The third step is data visualization, which visualizes the results and aids decision making.

Applications

Photogrammetry is a displacement measurement technology, which uses camera imaging to directly obtain the displacement value of the monitored objects and then processes the image data. It can be used to reconstruct 3D objects in either numerical or graphical form [83]. Laser scanners can reliably collect the geometric and radiation characteristics of the surrounding environment. When mounted on a high-precision mobile positioning device (such as UAV), the laser scanner can collect dense, accurate 3D point clouds at standard speeds [84]. UAVs are typically used to capture geographic images and generate a 3D grid of the terrain using imaging and photogrammetry techniques [83].

Construction Site Monitoring

Advanced technology is needed to monitor construction activities in real-time and visually update construction progress to improve the stability of the structure. Based on the combination of UAV and remote sensing system, a semiautomatic remote sensing non-destructive monitoring method was designed to monitor early concrete members and estimate their strength [85]. Teizer et al. [24] introduced the use of cameras and tracking algorithms to track multiple construction workers over a long period of time, identify their labour status and monitor unsafe behaviour. Taneja et al. [67] also introduced image capturing techniques such as laser scanners and cameras to collect accurate, complete and reliable field data.

Safety Inspection

Liu et al. [45] proposed a safety inspection method that combines UAV with BIM, including data collection and the construction of dynamic BIM model. Integrating the collected information with the BIM model enables the creation of a dynamic BIM model. This method does not only significantly improves the efficiency of safety inspections, but also enables managers in different locations to simultaneously watch video inspections and make timely decisions.

Structural Health Monitoring

Based on closed-circuit television (CCTV) video and deep learning algorithms, Yin et al. [86] developed a real-time automatic defect inspection system for sewer pipes, compared the defect inspection with a traditional artificial eye, and based on CV technology to help assess the automation of operation, increase productivity and help reduce the human error in CCTV video assessment operation. Alhaddad et al. [25] developed a new photogrammetry system based on digital image correlation technology to monitor tunnel deformation with an accuracy greater than 0.1 mm. Laser scanning technology is also now widely used in tunnel infrastructure measurement and monitoring. Puente et al. [38] presented a method to estimate tunnel vertical clearance from moving LIDAR point clouds. Mobile LiDAR technology can collect a large amount of accurate 3D information, reducing survey time and the risk of using traditional survey tools.

4.5.3. Geographic Information System

GIS is a spatial database management system. With the support of computer hardware and software systems, geospatial data can be captured, stored, processed, analyzed and displayed [3]. The advantages of GIS over other map management tools are data visualization capabilities, mathematical and statistical capabilities, especially the ability to make decisions using terrain [87]. GIS is an effective technology for civil infrastructure planning and maintenance. It provides an effective scheme for the maintenance of road databases, which serves as a foundation for road decision-making and road surface management. In practical applications, GIS is usually combined with GPS, AR, VR and other sensing technologies.

Applications

Zagvozda et al. [87] summarized the integration of GIS and pavement management systems, and introduced the evaluation and comparison of pavement conditions. By integrating sensor technology, spatial information technologies, 3D visualization technology and landslide prediction model, Huang et al. [88] developed a Web3DGIS system, which is used for real-time monitoring and early warning of landslides. Afferden et al. [89] discovered savings of up to 40% by using a GIS-based approach to evaluate local lowest-cost wastewater solutions. Chen et al. [4] used GIS to map suitable e-waste landfills. Wang et al. [3] proposed that the integration of GIS and geographic databases, AR and VR technologies could generate utility-based geospatial data for monitoring, recording and managing the location of all underground utilities. In addition, GIS is often used in conjunction with the GPS for positioning and mapping [90].

### 4.5.4. Micro-Electro-Mechanical System

MEMS is a small integrated device or system that combines computing and communication functions with sensing technologies [1]. MEMS devices are generally classified into three categories, including sensors, actuators and passive structures [91]. MEMS is developed based on a semiconductor manufacturing technology. The majority of MEMS applications in civil infrastructure field include MEMS accelerometers, MEMS optical sensors, MEMS temperature sensors, MEMS moisture sensors and MEMS pressure sensors, which can measure acceleration, straining, temperature, moisture and pressure.

Advantages

MEMS sensors have beneficial characteristics over traditional types of sensors, such as low cost and smaller size for mass production. MEMS sensors consume less power, are more sensitive to input changes, are less intrusive on the structure and last longer than other large monitoring devices.

Applications

Conventional strain gauges and FOS can be damaged in service due to the formation of cracks in reinforced concrete or slip between steel and concrete. Tondolo et al. [92] proposed a new scheme to monitor axial strain by combining embedded MEMS sensors to measure pressure and temperature changes. Moreover, MEMS embedded in concrete can help engineers monitor the solidification of concrete more effectively.

The MEMS integrated into WSN aids in low-cost, manageable monitoring of transportation infrastructure systems. Some scholars have developed an embedded wireless sensor platform, a small volume sensor device embedded in concrete to obtain sensor readings to monitor the state of the bridge deck, using wireless sensor network technology [91].

### 5. Discussions

*5.1. Application of Information Technologies in Different Phases*

Generally, the whole life cycle of a construction project can be divided into three phases, namely, planning and design, construction and maintenance. This study does not introduce in detail all the information technologies applications in civil infrastructure due to space constraints but instead classifies information technologies according to their application in different phases, as shown in Table 5.

**Table 5.** Information technologies are classified based on different phases of applications.

| Planning and Design | Construction | Maintenance |
| --- | --- | --- |
| 1. GIS<br><br>(Site selection [3,4,89], roadside vegetation management [93], and road investigation [90])<br><br>2. BIM<br>(Aid decision making [37], information sharing [47], collaborative design [111], 3D visualization [97] and LCM [39,98])<br>3. GNSS, GIS, AR and VR<br><br>(Visualization of underground facilities [117], investigation of road [90])<br><br>4. CV<br>(Planning [7,121], investigation of road [90])<br>5. GPR<br>(Exploration of underground structure [125,126], investigation of road [90]) | 1. BIM<br>(Information sharing [47], construction management optimization [47,94], schedule monitoring [5], digital delivery [95,96], 3D visualization [97] and LCM [39,98])<br>2. RFID<br><br>(Proximity sensing and safety warning [9,10], data collection and material tracking [66–68,112])<br><br>3. CV<br>(Data collection [24,108], visual inspection [38,84,112,118], quality control, schedule monitoring [5–8], efficiency improvement [119] and SHM [85])<br>4. GPS<br>(safety warning [122], material tracking [66] and site management [123])<br>5. System information model<br><br>(Schedule monitoring [127])<br><br>6. FOS<br><br>(Strain monitoring [58])<br><br>7. ICT, AI<br>(Schedule monitoring [132], site management [123] and communication [133])<br>8. Satellite remote sensing<br><br>(Schedule monitoring [141]) | 1. WSN<br><br>(SHM [11–13,18,19,21,23,27,30,52,70,99–106], information collection [107–109], quality monitoring [28] and traffic trend forecast [110])<br><br>2. FOS<br><br>(SHM [14–17,20,51,53–65,113–116])<br><br>3. BIM<br><br>(Expansion, update and maintenance [34,120], 3D visualization [36], safety inspection [45], SHM [35], asset management [96] and LCM [39,98])<br><br>4. RFID<br>(Leakage monitoring [70], object location [66,68,69] and proximity sensing [124])<br>5. Smartphone sensor<br>(Road detection [76–78], track wear assessment [128], traffic condition detection [79] and vibration monitoring [129])<br>6. MEMS<br>(Assessment performance and condition [130], SHM [91,92,131])<br>7. IoT and smart sensors<br><br>(SHM [25,73–75,81,116,134–140], information collection [107])<br><br>8. CV<br>(Data collection [83,142], visual inspection [7,38,84,86,143–146], SHM [25,147–149], safety inspection [8,45,150], road extraction [151] and pipeline positioning [152])<br>9. 3D printing<br>(Road repair [153])<br>10. AI, ML<br>(Passenger flow forecast [154], infrastructure assessment [155])<br>11. DAS<br>(Earthquake monitoring [156])<br>12. Big data<br>(Asset management [157], maintenance of railway condition [158])<br>13. GPR<br>(Exploration of underground structure [125], SHM [159–161])<br>14. GPS<br>(Vibration monitoring [162])<br>15. GIS<br>(SHM [163], pavement Management [87] and environmental impact assessment [164])<br>16. Remote sensing<br>(Environmental impact assessment [164])<br>17. Vehicular sensor networks<br>(Preventing rear-end collisions [165]) |

The following information can be obtained from Table 5:

Firstly, the application of information technologies in the planning and design phase is primarily in surveying site selection and assisting decision making. The application of information technologies in the construction phase mainly includes schedule and quality monitoring, data collection and material tracking, safety warning and construction site visual management. In the maintenance phase, the application of information technologies mainly includes SHM, condition assessment and asset management.

Secondly, in terms of the quantity, the quantity and application of information technologies in the planning and design phase is currently the least, followed by the construction phase. The maintenance phase employs and applies most information technologies.

### 5.2. Status of Research and Potential Future Research Directions

Based on the previous chapter's analysis of the application status and limitations of the three research phases, the current research framework and future research directions can be described. Given the limitations of the existing research, the potential research direction in the future is proposed Figure 7 depicts the current state of research and potential future research directions.

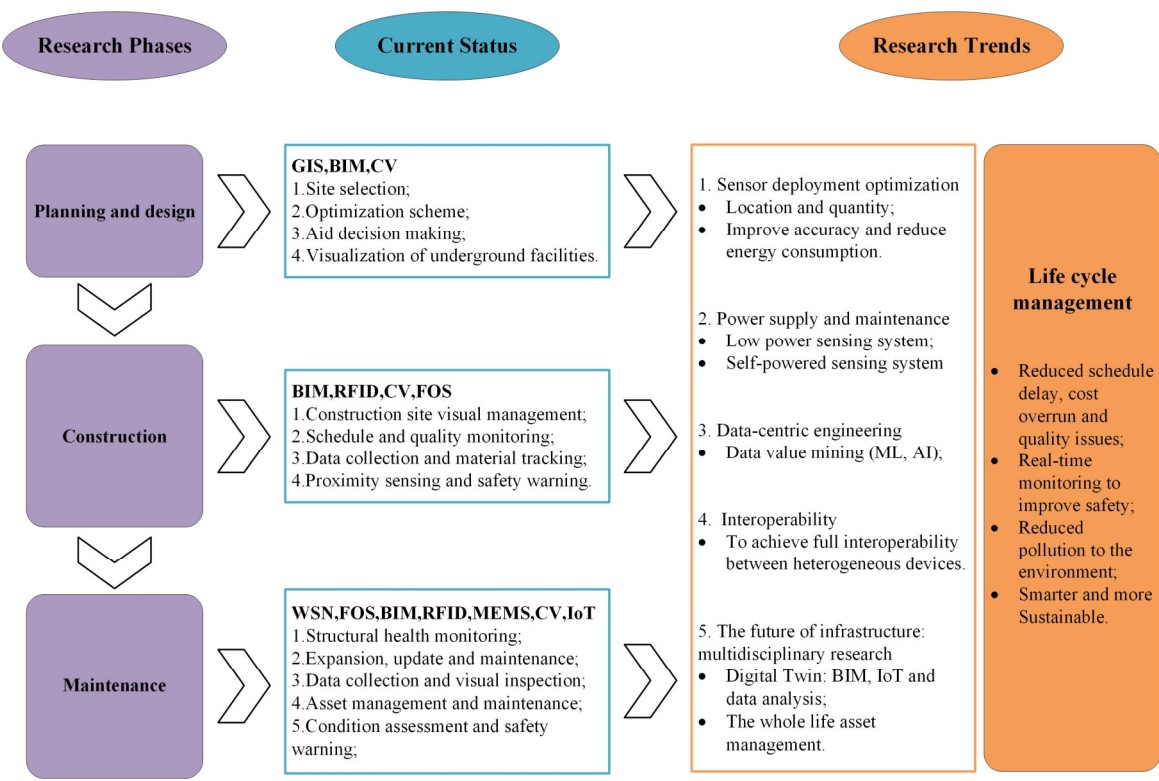

**Figure 7.** A framework of the status of research and potential future research directions.

### 5.2.1. Sensor Deployment Optimization

The optimal deployment of sensors generally depends on two factors, the location and number of sensors. The number of sensors determines the quantity and completeness of the sensor to collect data. If the selected sensors are insufficient, the data collected will be incomplete, potentially leading to wrong decisions. The location of the sensor determines the efficiency of the sensor to collect data. If the selected sensor is poorly located, the information gathered will limit the system's ability to detect and locate damage, reducing the lifespan of the sensing system [166].

The optimal deployment of sensors can improve data accuracy and reduce energy consumption. Although several scholars simulated the optimal deployment of sensors in the laboratory, the practical difficulties of deploying sensors in the field, including the actual wireless interference environment and data transmission efficiency, cannot be foreseen [31].

### 5.2.2. Power Supply and Maintenance

Powering a sensor system to monitor the dynamic performance of our infrastructure over the long term is one of the greatest challenges in civil infrastructure monitoring. Common sensing technologies, such as WSN, require a permanent power supply, which

may not be readily or easily available to all devices, high capacity batteries may be expensive and difficult to maintain [167].

An active research direction is the development of energy-saving sensor systems for the long-term monitoring of civil infrastructure. Sensor damage leads to a loss of data accuracy due to the large gap between the design life of the civil infrastructure and the sensor system, meanwhile, sensor update and maintenance is a challenge in civil infrastructure monitoring.

### 5.2.3. Data-Centric Engineering

Data are central to decision making, and high-quality data serve as the foundation for better decisions. Information sensing devices generate a large amount of data, but how is the valuable information extracted from the data identified? Applying advanced statistical methods, including machine learning and artificial intelligence techniques, to infrastructure is an emerging field of research [154]. To improve infrastructure management, high-level real-time statistical analysis based on collected data is required.

### 5.2.4. To Achieve Full Interoperability

Infrastructure asset data sets are stored on different platforms because these data sets are typically generated by multiple software applications using different standards and format types. As a result, the best value of data is frequently not fully realized without complete interoperability. Several relevant scholars have proposed that interoperability is a major obstacle to the seamless transmission of information between heterogeneous devices, and that ontology is the most promising approach to solving interoperability [50]. However, interoperability across disciplines remains a challenge in civil infrastructure management.

### 5.2.5. Future of Civil Infrastructure: Multidisciplinary Research

Intelligent systems for modern civil infrastructure face complex challenges that span many traditional disciplines, such as civil engineers, information technology engineers and data scientists. The digital twin, a cyber–physical system, can become a tool for the planning and management of next-generation smart infrastructure. The vision for a digital twin is to integrate artificial intelligence, machine learning and data analytics to create a digital simulation model that can continuously learn and self-update to represent near-real-time status, working conditions and the location of physical assets. It will be supported by BIM, data from the facilities management systems, IoT sensors and advanced data analytics.

In addition, a digital twin can reflect the full life cycle of the corresponding physical facility. For instance, the same data sets used in the design and construction phase can be used within the maintenance phase, allowing for new, innovative ways to apply the data. The digital twin will exist alongside the physical asset and will be updated through a statistical process as new data are collected to improve the life cycle management of the civil infrastructure.

## 6. Conclusions

In recent years, the application trends of information technologies in the civil infrastructure domain have grown rapidly. The purpose of this study is to determine the current state of development in this field and the outlook for future research direction.

The study collected 204 papers published in the past decade as literature samples to obtain the research trend. Then, VOSviewer software was used for statistical and visual analysis to determine the journals, highly cited papers and types of information technologies with more publications in this field. The information technologies involved in this paper mainly include FOS, BIM, WSN, RFID, CV and other advanced information sensing technologies. The authors expounded on the advantages, limitations and application status of these information technologies.

The application status and limitations of information technologies are determined based on the application of information technologies in different phases of the project life

cycle. In the planning and design phase, the application of information technologies is primarily in surveying site selection and assisting decision making. In the construction phase, the application of information technologies mainly includes schedule and quality monitoring, data collection and material tracking, safety warning and construction site visual management. The application of information technologies in the maintenance phase mainly includes SHM, condition assessment and asset management.

The authors anticipate future research directions such as sensor deployment optimization, power supply and maintenance, and interoperability between heterogeneous devices. Finally, the authors emphasize the importance of the whole life cycle management, arguing that a digital twin combined with multidisciplinary fields can become a tool for the planning and management of next-generation smart infrastructure, making the future of civil infrastructure smarter and more sustainable.

However, the literature samples selected in this study include English papers published on Web of Science, so some papers may not have been collected. Furthermore, this study analyzes these information technologies from the perspective of management and application, rather than focusing on specific technical principles. This research helps scholars to thoroughly understand the current status of information technologies application in civil infrastructure and allows them to build on previous research findings.

**Author Contributions:** Conceptualization, C.Z.L.; methodology, C.Z.L.; software, Z.G.; validation, C.Z.L.; formal analysis, Z.G. data curation, Z.G.; writing—original draft preparation, Z.G.; writing—review and editing, B.X. and V.W.Y.T.; visualization, D.S.; supervision, D.S. All authors have read and agreed to the published version of the manuscript.

**Funding:** This research was funded by the National Key R&D Program of China, grant number 2018YFB2100901; the National Natural Science Foundation of Guangdong Province [Grant No. 2021A1515012204 and 2021A1515110474]; the Department of Education of Guangdong Province [Grant No. 2021ZDZX1004]; and the Shenzhen Science and Technology Innovation Commission [Grant No. JCYJ20190808174409266, No. GJHZ20200731095806017 and No. SGDX20201103093600002].

**Institutional Review Board Statement:** Not applicable.

**Informed Consent Statement:** Not applicable.

**Data Availability Statement:** Not applicable.

**Acknowledgments:** All individuals included in this section have consented to the acknowledgement.

**Conflicts of Interest:** The authors declare that they have no known competing financial interest or personal relationships that could have appeared to influence the work reported in this paper.

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
