# Peer review of "The Application of Advanced Information Technologies in Civil Infrastructure Construction and Maintenance"

_sustainability, doi:10.3390/su14137761_

Round 1

Reviewer 1 Report

Overall, the manuscript presents a good review of recent articles on the topic of advanced information technologies. But the focus or scope appears too broad. A review paper can, and has been done on each of the methodologies discussed, e.g., building information modelling (BIM). Some other specific comments are as follows:

There is a lot of irrelevant initial discussion on some information such as the research universities, etc.

The manuscript should mention that the focus is on civil infrastructure, and not the general infrastructure from vertical and horizontal construction.

The grammar needs a thorough revision to enable good reading and comprehension.

Reviewer 2 Report

In this paper, the authors review the literature regarding the research development of information technologies used in civil infrastructure. The paper is well presented and adequately describes the objective, methodology and results.

More information regarding search process should be provided.

  • Why have you only considered Web of Science Core Collection?
  • What's the Eligibility Criteria (inclusion and exclusion criteria) of the paper’s selection? There is a considerable reduction of papers.
  • Why have they not been considered conference papers?
  • Has any unification of terms been carried out?

Tt is recommended to include an outline to reinforce the explanation and understanding of the methodology.

As possible, it is recommended to improve the quality of figure 1

The proposal sounds interesting and can have great potential to the readers of the journal to know directions for the future growth on this research topic.

Round 2

Reviewer 1 Report

The authors have made significant revisions to address the prior review. But the issues still remain that this review paper is not deep enough because the focus involves several important topics, in which of these topics (BIM, wireless sensors, etc.) can have review papers written on each of them. There are also still English grammar issues at several sections of the paper.

Reviewer 2 Report

The authors have clarified the issues raised in the major revision. 
